# Relationship between Oral Function and Support/Care-Need Certification in Japanese Older People Aged ≥ 75 Years: A Three-Year Cohort Study

**DOI:** 10.3390/ijerph192416959

**Published:** 2022-12-16

**Authors:** Komei Iwai, Tetsuji Azuma, Takatoshi Yonenaga, Taketsugu Nomura, Iwane Sugiura, Yujo Inagawa, Yusuke Matsumoto, Seiji Nakashima, Yoshikazu Abe, Takaaki Tomofuji

**Affiliations:** 1Department of Community Oral Health, School of Dentistry, Asahi University, 1851 Hozumi, Mizuho 501-0296, Japan; 2Gifu Dental Association, 1-18 Minamidori, Kano-cho, Gifu 500-8486, Japan

**Keywords:** chewing state, longitudinal study, older people, swallowing function, support/care-need certification

## Abstract

The aim was to examine the relationships between oral functions and support/care-need certification in older people aged ≥ 75 years using the National Health Insurance (NHI) database system and data from Kani City, Gifu, Japan. In total, 732 older Japanese people aged ≥ 75 years who did not have support/care-need certification and underwent dental check-ups in Kani City in 2017 were followed up until 2020. Chewing state, tongue and lip function, and swallowing function were assessed by a self-administered questionnaire, an oral diadochokinesis test, and a repetitive saliva-swallowing test, respectively. The presence or absence of systemic diseases and of support/care-need certification was based on data collected by the NHI database. At follow up, 121 (17%) participants had support/care-need certification. The participants with support/care-need certification included more women (*p* < 0.001) and older people (*p* < 0.001); and had more hypertension (*p* = 0.003), musculoskeletal disorders (*p* < 0.001), pneumonia (*p* = 0.044), poor chewing state (*p* < 0.001), and poor swallowing function (*p* = 0.003) than those without support/care-need certification. Furthermore, the presence of support/care-need certification at follow up was associated with sex (woman: odds ratio [OR], 2.120; 95% confidence interval [CI], 1.354 to 3.317), age (OR, 1.203; CI, 1.139 to 1.270), chewing state (poor: OR, 2.534; CI, 1.409 to 4.557), and swallowing function (poor: OR, 2.372; CI, 1.248 to 4.510) at baseline. However, tongue and lip function were not associated with support/care-need certification. The results indicate that older Japanese people aged ≥ 75 years with a poor chewing state and poor swallowing function at baseline had a higher risk for support/care-need certification after three years.

## 1. Introduction

As the global population continues to grow, having reached about 7.8 billion people as of 2020 [1], the population of older people is increasing remarkably, with more than 270 million people aged 75 years and over [2]. Against this backdrop, Japan is aging more rapidly than any other country, and as of 2020, the number of people aged 75 years and over was about 18.5 million, accounting for nearly 15% of the total population; in addition, it is expected to reach approximately 27% by 2060 [3,4,5]. An important issue in an aging society is the increasing prevalence of physical and mental disabilities with age [6,7].

In Japan, based on the long-term care insurance law, older people who need daily and continuous support or care due to various disabilities can receive seven levels of certifications, support-need certification 1–2, and care-need certification 1–5, according to their level of need for long-term care services, in descending order from those with mild symptoms [8]. In recent years, the proportion of older people with support/care-need certification has been increasing, with a report in 2020 showing that 32.5% of older people aged ≥ 75 years had support/care-need certification, which has become one cause of pressure on social security costs in Japan [9]. Therefore, it is very important to investigate the factors associated with support/care-need certification, and to explore immediate countermeasures in Japan.

Studies have reported the relationship between support/care-need certification and oral function. For example, it has been demonstrated that a significantly higher proportion of people with care-need certification had poor swallowing function compared to those without care-need certification [10,11,12]. It is also known that older people residing in geriatric health care facilities had a worse chewing state and tongue function than older people not residing in geriatric health care facilities [13,14]. However, since these observations are all based on cross-sectional studies, it is still unclear whether poor oral function contributes to support/care-need certification in older people. A longitudinal study reported that an oral frailty state in older people was a strong predictor of a long-term care condition in the future [15]. However, few studies have investigated the longitudinal relationships between poor oral functions and support/care-need certification, and further studies are needed.

In Japan, the National Health Insurance (NHI; also known as “Kokuho”) database system provides a database of the state of people with support/care-need certification. In addition, municipalities conduct dental check-ups of older people aged ≥ 75 years as part of public dental health services. This dental check-up includes items related to oral function, and it is expected that, by matching dental check-up data with the NHI database, a longitudinal study of oral function and the need for care can be conducted. However, few municipalities have a database of dental health check-up results, and there is no precedent for matching such data with the NHI database.

In the present study, we hypothesized that reduced oral function on dental health check-ups of older people aged ≥ 75 years conducted in municipalities would predict certification of the need for support and care in the NHI database. Kani City, Gifu Prefecture, is one of the municipalities that has a database of dental health check-up results. Therefore, the present study was a longitudinal study over a period of three years in which the aim was to clarify the longitudinal relationships between oral functions and support/care-need certification in Japanese older people aged ≥ 75 years using the NHI database and data from Kani City.

## 2. Materials and Methods

### 2.1. Participants

This study was designed as a 3-year cohort study. Data from community residents who underwent dental check-ups in Kani City, Gifu, Japan, were analyzed. The data of Kani city was provided on CD by the Gifu National Health Insurance Organization. Between April 2017 and March 2018, a total of 1171 older Japanese people aged ≥ 75 years participated in the baseline survey. The following were excluded from the analysis: participants with a medical history of dementia (66 participants), because they may not have been able to accurately understand functional testing instructions; participants with support/care-need certification at baseline (186 participants); participants with missing data for oral findings (67 participants); and participants with missing data for oral functions, including participants with missing data for chewing state (26 participants), tongue and lip function (15 participants), and swallowing function (32 participants). Of these 779 participants, 732 were followed up from April 2020 to March 2021 (follow-up rate, 94%). Therefore, the data of 732 community residents (399 men and 333 women, mean age 79.1 years) were analyzed in the present study (Figure 1).

### 2.2. Survey Items in the National Health Insurance Database System

Information about sex, age, and the presence or absence of hypertension, diabetes mellitus, dyslipidemia, musculoskeletal disorders, pneumonia, and support/care-need certification was obtained from the NHI database. In this study, “Certification” refers to all the people requiring support and long-term care due to restricted motor functions, as well as due to impaired thinking and comprehension.

### 2.3. Oral Items

Data from dental check-ups organized by Kani City, conducted in accordance with the manual for dental check-ups for older people aged ≥ 75 years recommended by the Ministry of Health, Labor and Welfare, were provided. Data for the following oral items were obtained: number of teeth present, presence or absence of periodontal pockets ≥ 4 mm, chewing state, tongue and lip function, and swallowing function. The coded values of the community periodontal index were used to evaluate the periodontal condition, with codes ≥1 being evaluated as periodontal pockets ≥ 4 mm [16]. The questionnaire items in chewing state included: “It is harder to eat hard food than it was six months ago (presence or absence)”. Participants who answered “presence” were defined as poor chewing state [17]. On the oral diadochokinesis test for tongue and lip function, “poor tongue and lip function” was defined as less than 30 syllables in 5 s of any one of “Pa”, “Ta”, or “Ka” [18]. For swallowing function, those who swallowed less than 3 times in 30 s in the repetitive saliva-swallowing test were evaluated as having poor swallowing function [19].

### 2.4. Statistical Analysis

The normality of the data was confirmed using the Shapiro-Wilk test. Because all the continuous variables were not normally distributed, data are expressed as medians (first and third quartiles). Significant differences in characteristics of the participants at baseline and at follow up were assessed using Fisher’s exact test or Wilcoxon’s signed-rank sum test. Significant differences in characteristics between the presence and absence of support/care-need certification were assessed using the chi-squared test and the Mann–Whitney U test. Univariate and multivariate logistic regression analyses were performed with the presence of support/care-need certification as the dependent variable. On multivariate stepwise logistic regression analysis, variables with *p* > 0.10 were excluded from the model; in addition, variables that were significantly different on univariate logistic regression analysis, in addition to sex and age, were selected for the third category. The suitability of this model was confirmed by the Hosmer–Lemeshow fit test. All the data were analyzed using statistical analysis software (SPSS statistics version 27; IBM Japan, Tokyo, Japan). All *p*-values < 0.05 were considered significant.

### 2.5. Research Ethics

The present study was approved by the Ethics Committee of Asahi University (No. 33006) and was performed in accordance with the Declaration of Helsinki (as revised in Brazil 2013). Informed consent was obtained from all the participants.

## 3. Results

Table 1 shows the characteristics of the participants at baseline and at follow up. Overall, there were 399 men (55%) and 333 women (45%). The proportion of participants with periodontal pockets ≥4 mm was significantly higher at follow up than at baseline (*p* = 0.043); there were no significant differences in the other factors between baseline and follow up.

Table 2 shows the differences in characteristics of participants at baseline with and without support/care-need certification at follow-up. In the present study, 121 participants (17%) were newly support/care-need certified at follow up. Participants with support/care-need certification were characterized by a significantly higher proportion of women (*p* < 0.001) and older (*p* < 0.001) than those without support/care-need certification. In addition, participants with support/care-need certification were characterized by a significantly higher proportion of those with hypertension (*p* = 0.003), musculoskeletal disorders (*p* < 0.001), pneumonia (*p* = 0.044), poor chewing state (*p* < 0.001), and poor swallowing function (*p* = 0.003), compared to those without support/care-need certification. On the other hand, there was no significant difference in tongue and lip function between participants with and without support/care-need certification in a three-year longitudinal study.

Table 3 shows crude odds ratios and 95% CIs for support/care-need certification at follow up. The results showed that the risk of support/care-need certification after three years was significantly associated with sex (woman: OR, 2.326; 95% CI, 1.556 to 3.478), age (OR, 1.233; 95% CI, 1.171 to 1.297), hypertension (yes: OR, 1.872; 95% CI, 1.236 to 2.837), musculoskeletal disorders (yes: OR, 2.535; 95% CI, 1.510 to 4.258), pneumonia (yes: OR, 1.542; 95% CI, 1.009 to 2.358), chewing state (poor: OR, 4.862; 95% CI, 2.935 to 8.054), and swallowing function (poor: OR, 2.257; 95% CI, 1.303 to 3.911) at baseline. On the other hand, there was no significant association between support/care-need certification after three years, and tongue and lip function at baseline.

Table 4 shows the adjusted odds ratios and 95% CIs for support/care-need certification at follow up. After adjusting for hypertension, musculoskeletal disorders, and pneumonia, the risk of support/care-need certification at three years was significantly associated with sex (woman: OR, 2.120; 95% CI, 1.354 to 3.317), age (OR, 1.203; 95% CI, 1.139 to 1.270), chewing state (poor: OR, 2.534; 95% CI, 1.409 to 4.557), and swallowing function (poor: OR, 2.372; 95% CI, 1.248 to 4.510) at baseline.

## 4. Discussion

To the best of our knowledge, this was the first study to examine longitudinal associations between oral function and support/care-need certification by matching dental check-up data and the NHI database in older Japanese people aged ≥ 75 years. The results showed that participants with support/care-need certification after three years had a higher proportion of poor chewing state and swallowing function than those without support/care-need certification. The results of logistic regression analysis showed that, after adjusting for sex, age, hypertension, musculoskeletal disorders, and pneumonia, the odds ratio of being support/care-need certification after three years showed higher with poor chewing state and poor swallowing function at baseline. From these results, it was predicted that poor chewing state and poor swallowing function had a higher risk of support/care-need certification in the future. In Japan, municipalities conduct annual dental check-ups for older people aged ≥ 75 years. The results of the present study suggest that the dental risk factors for certification of the need for care and support can be elucidated with content appropriate for each individual municipality by matching them with the NHI database. Having a database of the results of dental health check-ups would be important from the perspective of effectively using dental health in health policy for the prevention of the need for care.

The relationship between swallowing function and support/care-need certification may be related to poor dietary intake and nutrition. Previous reports suggested that poor swallowing function leads to poor feeding, which is a risk factor for malnutrition [20,21]. In addition, poor swallowing function is strongly associated as a factor in nutritional disorders in older people with care-need certification [22]. Furthermore, older people with support/care-need certification were a significantly higher proportion of those who had poor nutrition compared to older people without support/care-need certification [23]. Therefore, it is possible that the participants with poor swallowing function had a low dietary intake and poor nutritional state, resulting in weight loss and reduced activities of daily living, contributing to support/care-need certification.

In the present study, chewing state was associated with support/care-need certification. Previous studies have also shown an association between chewing state and support/care-need certification, and our study supports the findings of these studies [15,24]. This relationship may also be related to poor dietary intake and nutrition, as is the relationship between swallowing function and support/care-need certification. As the chewing state declines, the number of foods that can be ingested declines, making the patient more susceptible to a low nutritional state [25]. In addition, a decline in oral function may lead to systemic frailty and support/care-need certification [15,26].

In the present study, tongue and lip function was not associated with support/care-need certification. Previous reports have shown associations between tongue function and care-need certification, which are different from the results of the present study [27]. The results in the previous and present studies may have been different due to different measurement methods. In previous studies, the evaluation of tongue function was based on the magnitude of tongue pressure. In the present study, the data for tongue and lip function were based on dental check-ups, which were conducted in accordance with the manual for dental check-ups for older people aged ≥ 75 years. The present study was based on dental health examination data for screening purposes and may include false positives and false negatives for actual tongue- and lip-function decline. Therefore, although the degrees of false positives and false negatives of dental check-ups is currently unknown, it may have been better if additional examinations of actual tongue and lip function had been performed.

A major strength of the present study is that it included a sample size of more than 700 older Japanese people aged ≥ 75 years. Furthermore, this was a longitudinal study, which is useful for establishing a causal relationship between support/care-need certification and poor chewing state, or swallowing function; and for inferring factors that contribute to the independent living of older people.

However, there are several limitations to the present study. First, the exercise habits and social environment (education and income) of participants during the study period were not taken into account [28,29]. They should be investigated in order to develop municipal strategies to increase the number of older people living independently. Second, nutritional state was not assessed in the present study. Third, since the participants of the present study had participated in dental check-ups, they may be a highly health-conscious population. Finally, our study utilizes participants’ responses to a self-reported questionnaire to assess chewing state. A previous study showed that self-reported chewing state in specific medical check-ups was related to the number of decayed teeth and periodontal pockets, based on Eichner’s classification [30]. It has also been reported that chewing state confirmed by a self-reported questionnaire is not only related to number of present, molar, and functional teeth, but is also useful as a screening method for actual chewing ability [31]. However, there may be a discrepancy between chewing awareness and actual chewing ability.

## 5. Conclusions

The present study matching dental check-up data with the NHI database showed that older Japanese people aged ≥ 75 years with a poor chewing state and poor swallowing function at baseline had a higher risk for support/care-need certification after three years. However, there was no significant association between tongue and lip function, and support/care-need certification.

## Figures and Tables

**Figure 1 ijerph-19-16959-f001:**
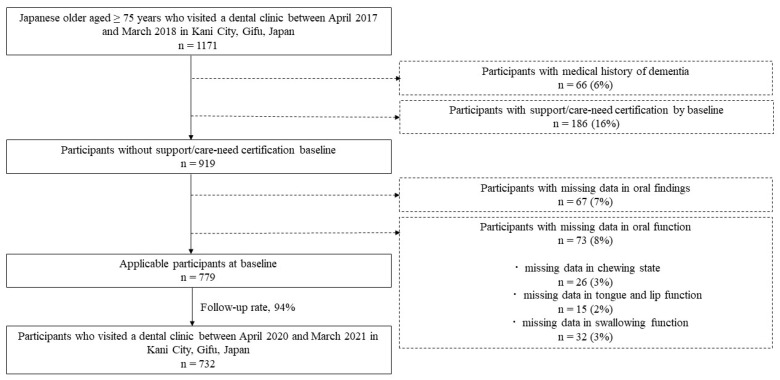
Flowchart of data selection criteria.

**Table 1 ijerph-19-16959-t001:** Participants’ characteristics (*n* = 732).

Factor	Baseline	Follow Up	*p*-Value *
Sex ^†^	399 (55%)	399 (55%)	-
Age (y)	78 (76, 81)	81 (79, 84)	-
Hypertension ^‡^	412 (56%)	419 (57%)	0.711
Diabetes mellitus ^‡^	280 (38%)	287 (39%)	0.788
Dyslipidemia ^‡^	353 (48%)	362 (50%)	0.637
Musculoskeletal disorders ^‡^	517 (71%)	547 (75%)	0.078
Pneumonia ^‡^	182 (25%)	193 (26%)	0.509
Number of present teeth	24 (19, 27)	24 (19, 27)	0.161
Periodontal pockets ≥ 4 mm ^‡^	456 (62%)	493 (67%)	0.042
Chewing state ^§^	77 (11%)	79 (11%)	0.865
Tongue and lip function ^§^	11 (2%)	17 (2%)	0.331
Swallowing function ^§^	73 (10%)	88 (12%)	0.275

Continuous variables are expressed as medians (first quartile, third quartile). ^†^: men, *n* (%); ^‡^: yes, *n* (%); ^§^: poor, *n* (%). * *p*  <  0.05, using Fisher’s exact test or Wilcoxon’s signed-rank sum test.

**Table 2 ijerph-19-16959-t002:** Differences in characteristics of participants at baseline with and without support/care-need certification at follow-up.

Factor	Support/Care-Need Certification at Follow Up		*p*-Value *
	Absent (*n* = 611)	Present (*n* = 121)	
Sex ^†^	354 (58%)	45 (37%)	<0.001
Age (y)	78 (76, 80)	81 (78, 85)	<0.001
Hypertension ^‡^	329 (54%)	83 (69%)	0.003
Diabetes mellitus ^‡^	228 (37%)	52 (43%)	0.242
Dyslipidemia ^‡^	290 (48%)	63 (52%)	0.355
Musculoskeletal disorders ^‡^	415 (68%)	102 (84%)	<0.001
Pneumonia ^‡^	144 (23%)	39 (32%)	0.044
Number of present teeth	24 (19, 27)	24 (20, 27)	0.337
Periodontal pockets ≥ 4 mm ^‡^	381 (63%)	75 (62%)	0.938
Chewing state ^§^	44 (7%)	33 (27%)	<0.001
Tongue and lip function ^§^	8 (1%)	3 (3%)	0.334
Swallowing function ^§^	52 (9%)	21 (17%)	0.003

Continuous variables are expressed as medians (first quartile, third quartile). ^†^: men, *n* (%); ^‡^: yes, *n* (%); ^§^: poor, *n* (%). * *p*  <  0.05, using Fisher’s exact test or the Mann–Whitney U test.

**Table 3 ijerph-19-16959-t003:** Crude odds ratios and 95% CIs for support/care-need certification at follow up.

Factor		ORs	95% CI	*p*-Value
Sex	Man	1	(reference)	<0.001
	Woman	2.326	1.556–3.478	
Age (y)		1.233	1.171–1.297	<0.001
Hypertension	No	1	(reference)	0.003
	Yes	1.872	1.236–2.837	
Diabetes mellitus	No	1	(reference)	0.243
	Yes	1.266	0.852–1.880	
Dyslipidemia	No	1	(reference)	0.355
	Yes	1.202	0.814–1.777	
Musculoskeletal disorders	No	1	(reference)	<0.001
	Yes	2.535	1.510–4.258	
Pneumonia	No	1	(reference)	0.045
	Yes	1.542	1.009–2.358	
Number of present teeth		0.987	0.960–1.014	0.337
Periodontal pockets ≥ 4 mm	No	1	(reference)	0.938
	Yes	0.984	0.659–1.471	
Chewing state	Well	1	(reference)	<0.001
	Poor	4.862	2.935–8.054	
Tongue and lip function	Well	1	(reference)	0.342
	Poor	1.916	0.501–7.329	
Swallowing function	Well	1	(reference)	0.004
	Poor	2.257	1.303–3.911	

Abbreviations: ORs, odds ratios; CI, confidence interval.

**Table 4 ijerph-19-16959-t004:** Adjusted odds ratios and 95% CIs for support/care-need certification at follow up.

Factor		ORs	95% CI	*p*-Value
Sex	Man	1	(reference)	0.001
	Woman	2.120	1.354–3.317	
Age (y)		1.203	1.139–1.270	<0.001
Hypertension	No	1	(reference)	0.189
	Yes	1.368	0.857–2.182	
Musculoskeletal disorders	No	1	(reference)	0.091
	Yes	1.646	0.924–2.932	
Pneumonia	No	1	(reference)	0.111
	Yes	1.470	0.915–2.361	
Chewing state	Well	1	(reference)	0.002
	Poor	2.534	1.409–4.557	
Swallowing function	Well	1	(reference)	0.008
	Poor	2.372	1.248–4.510	

Abbreviations: ORs, odds ratios; CI, confidence interval. Adjustment for sex, age, hypertension, musculoskeletal disorders, pneumonia, chewing state, and swallowing function. Hosmer-Lemeshow Fit test; *p* = 0.812.

## Data Availability

The data are not publicly available due to restrictions for reasons of privacy and ethics. The data presented in this study are available on request from the corresponding author.

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
