# Peer review of "Relationship between Oral Function and Support/Care-Need Certification in Japanese Older People Aged ≥ 75 Years: A Three-Year Cohort Study"

_ijerph, 2022, doi:10.3390/ijerph192416959_

Round 1
Reviewer 1 Report
This was an interesting study on the relationship of oral function with the likelihood for using the NHI.
Title: I find the title too long: Perhaps shorten it to "Relationship between Oral Function and Support/care-need Certification in Japanese Older People Aged ≥ 75 Years: A Three-year Cohort Study"
Abstract:
1) Line 15: The aim was to examine the relationships between oral functions and support/care-need 15 certification in 732 older people aged ≥ 75 years using the National Health Insurance database system (KDB) and data from Kani City, Gifu, Japan: Sample size should not be mentioned in the aim but part of the methods/results. It is also not suitable to have an abbreviation KDB that is shown early on next the a phrase that does not have the initials. For non-Japanese, I find it confusing that the authors use KDB instead of NHI. I would like to suggest that the authors use the abbreviation NHI throughout the text. However, if they wish to introduce the term Kokuho, use the example from their government's website: National Health Insurance (NHI; also known as "Kokuho") database.
2) Line 18: As oral functions, chewing state, tongue and lip function, and swallowing function were investigated: I don't understand what this sentence.
3) Line 23: At follow-up, 121 (17%) participants had support/care-need certification: You should state in the methods that the subjects included those without certification at the beginning.
4) Line 31: Please rephrase the final sentence to sound like a conclusion.
Introduction
1) Lines 38 and 42: In the first sentence, is the data referring to Japanese population or world population?
2) Line 47: are the values 1-2 and 1-5 referring to the types or level?
3) Line 67: This is an international journal. It is hard to remember a foreign word as a policy and I confuse myself thinking this is a location. As mentioned above, it would be less confusing to non-Japanese if the term National Health Insurance (NHI; also known as "Kokuho") database is used instead. Then you can refer later in the text as NHI database. Alternatively, you can state as National Health Insurance (also known as "Kokuho") database (NHID), and refer in the subsequent text as NHID. I prefer the former. as in line 97, the term National Health Insurance database was used as the heading. So I think you should be consistent with the term used.
4) Line 79: Wouldn't it be better to be more specific and refer this study as a cohort study?
Materials and methods
1) Line 84: Please cite the data source e.g. name of the government/institution body who kept the database in Kani City.
2) Line 110: Are both codes 1 and 2 have the same interpretation "periodontal pockets ≥ 4 mm"?
3) Line 111: The questionnaire items in chewing state included “It is harder to eat hard food than it was six months ago (presence or absence)”. Participants who answered "absence" were defined as poor chewing state: Shouldn't it be the opposite? having to agree (presence) to this statement, then the participant agreed that they found it hard to chew. So if they stated absent, they would have good chewing state.
Results
1) Table 2 title "Baseline characteristics of participants with support/care-need certification and participants without support/care-need certification at follow-u": the word should be "follow-up" i.e. missing letter p. secondly, I find it confusing for using the term baseline. I thought this referred to the time in 2017. but at follow up it is in 2020. I also found the values in brackets next to the numbers of absent or present confusing. I see the legend indicate explained for continuous variables and values in brackets referred to proportions. But what are the non continuous variables referring to? absolute number, n? I am not sure how the values within each row relate to the values of the columns. Could you make the table easier to understand?
Discussion
1) Line 182: The results of the present study suggest that the dental risk factors for certification of the need for care and support can be elucidated with content appropriate for each individual municipality by matching them with the KDB. It is desirable to have a database of the results of dental check-ups to match them with the KDB: I don't understand what the sentences were trying to suggest. Do you mean to suggest for the municipality to provide data of the dental risk factors to NHI? To what benefit would that do, would that then cause the NHI to impose higher charges to those with dental risk factors because they are likely to have the need for care? Perhaps knowing the authors could suggest to the government to increase prevention strategies that target those with poor chewing state and swallowing function because they're likely to be malnourished and have to get care later.
Conclusion
1) Line 240: Japanese older people aged ≥ 75 years with poor chewing state and poor swallowing function 241 have a higher risk for future support/care-need certification: Can you rephrase, Japanese older people aged ≥ 75 years with poor chewing state and poor swallowing function AT BASELINE have a higher risk for future support/care-need certification.
Author Response
This was an interesting study on the relationship of oral function with the likelihood for using the NHI.
Title: I find the title too long: Perhaps shorten it to "Relationship between Oral Function and Support/care-need Certification in Japanese Older People Aged ≥ 75 Years: A Three-year Cohort Study"
Our response: Thank you for your comment. As you suggested, we have changed the title to “Relationship between Oral Function and Support/care-need Certification in Japanese Older People Aged ≥ 75 Years: A Three-year Cohort Study”.
Abstract:
1) Line 15: The aim was to examine the relationships between oral functions and support/care-need certification in 732 older people aged ≥ 75 years using the National Health Insurance database system (KDB) and data from Kani City, Gifu, Japan: Sample size should not be mentioned in the aim but part of the methods/results. It is also not suitable to have an abbreviation KDB that is shown early on next the a phrase that does not have the initials. For non-Japanese, I find it confusing that the authors use KDB instead of NHI. I would like to suggest that the authors use the abbreviation NHI throughout the text. However, if they wish to introduce the term Kokuho, use the example from their government's website: National Health Insurance (NHI; also known as "Kokuho") database.
Our response: Thank you for your comments. We are agreement with your suggestions. We have revised the sentences to “The aim was to examine the relationships between oral functions and support/care-need certification in older people aged ≥ 75 years using the National Health Insurance (NHI) database system and data from Kani City, Gifu, Japan” (lines 14-16). Also, the text was changed from "KDB" to "NHI database" throughout (lines 15, 20, 62, 66, 68, 71, 75, 98, 178, 188, 239).
2) Line 18: As oral functions, chewing state, tongue and lip function, and swallowing function were investigated: I don't understand what this sentence.
Our response: Thank you for your comment. We have deleted this sentence.
3) Line 23: At follow-up, 121 (17%) participants had support/care-need certification: You should state in the methods that the subjects included those without certification at the beginning.
Our response: Thank you for your comment. We have added the words “who were not support/care-need certification” in the methods section (lines 16-17).
4) Line 31: Please rephrase the final sentence to sound like a conclusion.
Our response: Thank you for your comments. We have changed the sentence to “The results indicate that Japanese older people aged ≥ 75 years with poor chewing state and poor swallowing function at baseline had a higher risk for support/care-need certification after three years” (lines 28-30).
Introduction
1) Lines 38 and 42: In the first sentence, is the data referring to Japanese population or world population?
Our response: Thank you for your comment. In the first sentence, the data is referring to world population. We have changed the words from “the population” to “the global population” (line 35).
2) Line 47: are the values 1-2 and 1-5 referring to the types or level?
Our response: Thank you for your comment. The values 1-2 and 1-5 are referring to the levels. We have changed the word from “types” to “levels” (line 43).
3) Line 67: This is an international journal. It is hard to remember a foreign word as a policy and I confuse myself thinking this is a location. As mentioned above, it would be less confusing to non-Japanese if the term National Health Insurance (NHI; also known as "Kokuho") database is used instead. Then you can refer later in the text as NHI database. Alternatively, you can state as National Health Insurance (also known as "Kokuho") database (NHID), and refer in the subsequent text as NHID. I prefer the former. as in line 97, the term National Health Insurance database was used as the heading. So I think you should be consistent with the term used.
Our response: Thank you for your comments. We are agreement with your suggestions. The text was changed from "KDB" to "NHI database" throughout (lines 15, 20, 62, 66, 68, 71, 75, 98, 178, 188, 239).
4) Line 79: Wouldn't it be better to be more specific and refer this study as a cohort study?
Our response: Thank you for your comment. We have added a sentence “This study was designed as a 3-year cohort study” (line 79).
Materials and methods
1) Line 84: Please cite the data source e.g. name of the government/institution body who kept the database in Kani City.
Our response: Thank you for your comment. The data of Kani city was provided on CD by the Gifu National Health Insurance Organization. We have added the sentence (lines 80-81).
2) Line 110: Are both codes 1 and 2 have the same interpretation "periodontal pockets ≥ 4 mm"?
Our response: Thank you for your comment. Both codes 1 and 2 can be interpreted as "periodontal pocket ≥4 mm". To avoid misinterpretation, we have changed the text to read " The coded values of the Community Periodontal Index were used to evaluate periodontal condition, with codes ≥1 being evaluated as periodontal pockets ≥ 4 mm [16] " (lines 107-109).
3) Line 111: The questionnaire items in chewing state included “It is harder to eat hard food than it was six months ago (presence or absence)”. Participants who answered "absence" were defined as poor chewing state: Shouldn't it be the opposite? having to agree (presence) to this statement, then the participant agreed that they found it hard to chew. So if they stated absent, they would have good chewing state.
Our response: Thank you for your comments. The correct terminology is to define participants who answered "presence" as having poor chewing state. We have corrected the word from “absence” to "presence" (line 110).
Results
1) Table 2 title "Baseline characteristics of participants with support/care-need certification and participants without support/care-need certification at follow-u": the word should be "follow-up" i.e. missing letter p. secondly, I find it confusing for using the term baseline. I thought this referred to the time in 2017. but at follow up it is in 2020. I also found the values in brackets next to the numbers of absent or present confusing. I see the legend indicate explained for continuous variables and values in brackets referred to proportions. But what are the non continuous variables referring to? absolute number, n? I am not sure how the values within each row relate to the values of the columns. Could you make the table easier to understand?
Our response: Sorry for the confusing table. We have changed the table title to “Differences in characteristics of participants at baseline with and without support/care-need certification at follow-up”. In addition, non-continuous variables are expressed as n (%). We have revised Table 2. We have also revised the sentence (lines 143-144).
Discussion
1) Line 182: The results of the present study suggest that the dental risk factors for certification of the need for care and support can be elucidated with content appropriate for each individual municipality by matching them with the KDB. It is desirable to have a database of the results of dental check-ups to match them with the KDB: I don't understand what the sentences were trying to suggest. Do you mean to suggest for the municipality to provide data of the dental risk factors to NHI? To what benefit would that do, would that then cause the NHI to impose higher charges to those with dental risk factors because they are likely to have the need for care? Perhaps knowing the authors could suggest to the government to increase prevention strategies that target those with poor chewing state and swallowing function because they're likely to be malnourished and have to get care later.
Our response: Thank you for your comments. What we wanted to insist on was “Having a database of the results of dental health check-ups would be important from the perspective of effectively using dental health in health policy for the prevention of the need for care”. Therefore, we have deleted the sentences “It is desirable to have a database of the results of dental check-ups to match them with the NHI database. Unfortunately, few municipalities have a database of dental health check-up results”.
Conclusion
1) Line 240: Japanese older people aged ≥ 75 years with poor chewing state and poor swallowing function 241 have a higher risk for future support/care-need certification: Can you rephrase, Japanese older people aged ≥ 75 years with poor chewing state and poor swallowing function AT BASELINE have a higher risk for future support/care-need certification.
Our response: Thank you for your comment. We have revised the sentence to “The present study matching dental check-up data with the NHI database showed that Japanese older people aged ≥ 75 years with poor chewing state and poor swallowing function at baseline had a higher risk for support/care-need certification after three years”(lines 239-241).
Reviewer 2 Report
Normality can better be investigated by the Shapiro-Wilk test.
p.5 l.3 associated seems better than ``correlated’’ as the latter is not computed, see also elsewhere.
The somewhat vague expression ``was associated’’ is repeated quite often, whereas the risk interpretations of the ODDs ratios is lacking from the manuscript. Such would increase clarity of the findings. Making such interpretations would also sustain the final conclusion better.
p. 7 top: The discussed point about FP and FN is in principle justified, but it seems better to find literature to make this more specific and/or to add that the degree in which this occurs is currently unknown.
Author Response
Normality can better be investigated by the Shapiro-Wilk test.
Our response: Thank you for your comment. The normality of our data was re-confirmed using the Shapiro-Wilk test (line 117). The result that the distribution is not normal was the same as before test.
p.5 l.3 associated seems better than ``correlated’’ as the latter is not computed, see also elsewhere.
Our response: Thank you for your comment. We have revised the word from “correlated” to “associated” (lines 158, 169).
The somewhat vague expression ``was associated’’ is repeated quite often, whereas the risk interpretations of the ODDs ratios is lacking from the manuscript. Such would increase clarity of the findings. Making such interpretations would also sustain the final conclusion better.
Our response: Thank you for your comment. We have revised the sentence (182-185).
p. 7 top: The discussed point about FP and FN is in principle justified, but it seems better to find literature to make this more specific and/or to add that the degree in which this occurs is currently unknown.
Our response: Thank you for your comment. We have added the words “the degrees of false positives and false negatives of dental check-ups is currently unknown” (lines 216-217). We also have deleted the words “, which may result in false negatives or false positives” (line 236).